# Isolation and Characterization of the First *Zobellviridae* Family Bacteriophage Infecting *Klebsiella pneumoniae*

**DOI:** 10.3390/ijms24044038

**Published:** 2023-02-17

**Authors:** Roman B. Gorodnichev, Maria A. Kornienko, Maja V. Malakhova, Dmitry A. Bespiatykh, Valentin A. Manuvera, Oksana V. Selezneva, Vladimir A. Veselovsky, Dmitry V. Bagrov, Marina V. Zaychikova, Veronika A. Osnach, Anna V. Shabalina, Oleg V. Goloshchapov, Julia A. Bespyatykh, Anna S. Dolgova, Egor A. Shitikov

**Affiliations:** 1Lopukhin Federal Research and Clinical Center of Physical-Chemical Medicine of Federal Medical Biological Agency, 119435 Moscow, Russia; 2Department of Bioengineering, Faculty of Biology, Lomonosov Moscow State University, 119234 Moscow, Russia; 3Saint Petersburg Pasteur Institute, Federal Service on Consumer Rights Protection and Human Well-Being Surveillance, 197101 St. Petersburg, Russia; 4R.M. Gorbacheva Memorial Institute of Oncology, Hematology and Transplantation, Pavlov First Saint Petersburg State Medical University, 197022 St. Petersburg, Russia

**Keywords:** *Klebsiella pneumoniae*, multidrug resistance, capsular type, capsule depolymerase, bacteriophage, *Zobellviridae*

## Abstract

In order to address the upcoming crisis in the treatment of *Klebsiella pneumoniae* infections, caused by an increasing proportion of resistant isolates, new approaches to antimicrobial therapy must be developed. One approach would be to use (bacterio)phages and/or phage derivatives for therapy. In this study, we present a description of the first *K. pneumoniae* phage from the *Zobellviridae* family. The vB_KpnP_Klyazma podovirus, which forms translucent halos around the plaques, was isolated from river water. The phage genome is composed of 82 open reading frames, which are divided into two clusters located on opposite strands. Phylogenetic analysis revealed that the phage belongs to the *Zobellviridae* family, although its identity with the closest member of this family was not higher than 5%. The bacteriophage demonstrated lytic activity against all (*n* = 11) *K. pneumoniae* strains with the KL20 capsule type, but only the host strain was lysed effectively. The receptor-binding protein of the phage was identified as a polysaccharide depolymerase with a pectate lyase domain. The recombinant depolymerase protein showed concentration-dependent activity against all strains with the KL20 capsule type. The ability of a recombinant depolymerase to cleave bacterial capsular polysaccharides regardless of a phage’s ability to successfully infect a particular strain holds promise for the possibility of using depolymerases in antimicrobial therapy, even though they only make bacteria sensitive to environmental factors, rather than killing them directly.

## 1. Introduction

*Klebsiella pneumoniae* is a Gram-negative, non-motile, facultative anaerobic bacterium that occurs ubiquitously in nature and can be found in the normal flora of humans and animals [1,2]. At the same time, *K. pneumoniae* is a pathogen that causes up to 32% of all nosocomial infections [3], posing a serious challenge to treatment, particularly due to the rapid increase in the number of multidrug-resistant (MDR) strains [4]. According to the latest data, more than a third of *K. pneumoniae* isolates in Europe are resistant to at least one class of antibiotics: fluoroquinolones (33.6%), third-generation cephalosporins (34.3%), aminoglycosides (23.7%), and carbapenems (11.7%). Approximately a fifth (21.2%) are resistant to all the aforementioned classes [5]. Infections caused by resistant *K. pneumoniae* strains have a high mortality rate and rank third in the world in terms of number of deaths among infections caused by drug-resistant bacteria [6].

Moreover, since the mid-1980s, a special pathotype of hypervirulent *K. pneumoniae* (hvKp) has been distinguished. This pathotype is able to cause severe infections in immunocompromised people and in people who were healthy prior to infection, and can lead to subsequent intra-abdominal abscesses, which are prone to manifest in other tissues [7,8,9]. In general, the hvKp pathotype is considered a separate evolutionary lineage and is known for having a significantly lower number of MDR strains than classical *K. pneumoniae* (cKp). However, in recent years, there have been more reports of convergent *K. pneumoniae* clones that are both multidrug-resistant and hypervirulent [10,11].

For the treatment of infections caused by *K. pneumoniae*, phage therapy may be used as an alternative to antibiotic therapy [12]. Bacteriophages are the most abundant and widespread group of viruses that infect and replicate within bacteria, and are present in both natural and human microbiomes [13]. Due to their ability to infect and lyse bacterial cells, phages have been used as antimicrobials since their discovery in the early 20th century [14]. Today, monophages or customized cocktails of several lytic phages are successfully used in personalized therapy [15,16,17,18]; however, the efficacy of commercial broad-spectrum phage cocktails remains limited [19].

Besides the direct use of phages, the possibility of using antimicrobial bacteriophage-derived proteins such as endolysins and depolymerases is also being actively researched [20,21]. Endolysins are phage enzymes represented by peptidoglycan hydrolases that destroy the bacterial cell wall when a new phage population enters the environment [22]. Depolymerases are usually represented by structural proteins—fibers and spikes, which determine the adsorption of phage virions and thereby the host range [23]. Unlike endolysins, depolymerases do not lyse the bacterium, but cleave its capsular polysaccharides, significantly reducing the bacterium’s resistance to external environmental factors: antibiotics or the host’s immune system [24,25,26]. Usually, depolymerases have a narrow specificity limited by a specific type of bacterial capsular polysaccharide (CPS) [27].

In this work, we report the isolation and characterization of a novel lytic phage—vB_KpnP_Klyazma, which is the first *K. pneumoniae* phage from the *Zobellviridae* family. We also identified and obtained a recombinant phage-derived depolymerase protein, Kl-dep, that showed specificity against *K. pneumoniae* with the KL20 capsular type.

## 2. Results

### 2.1. Isolation and Phenotypic Characterization of vB_KpnP_Klyazma

*K. pneumoniae* strain L2-1B was isolated in 2020 from a patient with a chronic throat infection and was used as the host strain. The strain showed sensitivity to meropenem, cefotaxime, gentamicin, levofloxacin, tetracycline, and colistin and was resistant to erythromycin (Appendix A). Strain L2-1B belongs to the sequence type 268 (ST268) and has the capsular type KL20. The strain forms large mucoid colonies that, according to the string test, can be classified as hypervirulent. According to PCR analysis, the genome of the L2-1B strain carried virulence genes such as *iutA* and *fimH*, as well as plasmid variants of the *rmpA*, *rmpA2*, and *peg-344* genes, which are specific to hvKp [28].

The vB_KpnP_Klyazma phage was isolated from the Klyazma River in Korolev, Moscow oblast. The vB_KpnP_Klyazma phage plaques were clear and round, about 1–3 mm in diameter, with translucent halos, which is typical for phages carrying polysaccharide depolymerase genes (Figure 1A) [29].

Transmission electron microscopy (TEM) observation revealed that vB_KpnP_Klyazma has a symmetrical polyhedral head (~66 nm in diameter) and a short and non-contracting tail (~18 nm long) (Figure 1B), which are typical morphological characteristics of podoviruses [30].

The one-step growth curve showed that vB_KpnP_Klyazma had a latent period of 35 min followed by a rise period of 20 min (Appendix A). The vast majority of phage particles were adsorbed onto host strain L2-1B by 5–6 min of incubation (Appendix A). The burst size of the vB_KpnP_Klyazma phage was about 64 plaque-forming units per cell (PFU/cell).

Stability tests showed that the vB_KpnP_Klyazma phage is stable in the temperature range 4–55 °C and pH from 6 to 9. Incubation at temperatures of 65 °C and 75 °C or pH 4–5 and 10 reduced the bacteriophage titer by several orders of magnitude. Incubation at a higher temperature (>55 °C) or pH < 4 or >10 rapidly reduced the phage titer to zero (Appendix A).

### 2.2. Genetic Organization and Phylogenetic Affiliation of vB_KpnP_Klyazma

vB_KpnP_Klyazma has a linear double-stranded DNA genome of 50,298 bp with a G+C content of 46% and 711 bp long terminal repeats present at both termini. The sequence analysis resulted in assigning 82 putative open reading frames (ORFs) with a total length of 46,362 bp (92.2% encoding percentage).

The phage’s genes were categorized into two main modules: “structure and morphogenesis” on the plus strand and “replication, regulation, transcription, and translation” on the minus strand. The cluster of structure and morphogenesis contains 24 ORFs, 11 of which encode typical structural proteins of T7-like podoviruses: tail fiber protein (orf001), terminase large subunit (orf004), putative portal protein (orf005), putative scaffold protein, major capsid protein (orf007 and orf008), tail tubular proteins A and B (orf010 and orf012), and putative internal virion proteins A, B, D, and C (orf013, orf014, orf015, and orf016). The vB_KpnP_Klyazma genome does not carry genes encoding tRNA, RNA polymerase, or integrases, nor known determinants of antibiotic resistance or virulence, or toxins.

Analysis of the tail fiber protein (orf001) via HHPred search against the Pfam-A_v35 database revealed a parallel beta helix structure as well as a Pectate_lyase_3 domain at position 208–459 aa. This is a characteristic feature of polysaccharide depolymerases. There are three *K. pneumoniae* phages that are known to carry depolymerases specific to this capsular type: *Klebsiella* phage KpV766 (GenBank accession no. KX712071.1), *Klebsiella* phage KpV289 (GenBank accession no. NC_028977.1), and *Klebsiella* phage vB_KpnM_VIK251 (GenBank accession no. MZ602147.1) [31,32,33]. The KpV766 and KpV289 phages belong to the family *Autographiviridae* and the genus *Przondovirus*, while vB_KpnM_VIK251 belongs to the subfamily *Vequintavirinae* and the genus *Mydovirus*. Despite the differences in the phylogenetic position of the vB_KpnP_Klyazma, vB_KpnM_VIK251, KpV766, and KpV289 phages, their receptor-binding proteins (RBPs) are represented by depolymerases with a pectate lyase domain; however, analysis using BLASTp only showed homology between the RBPs of the KpV766 and KpV289 phages (100% query coverage and 94.80% identity).

BLASTn analysis showed that the vB_KpnP_Klyazma phage is closest to the *Zobellviridae* family phage *Citrobacter* phage CVT22 (GenBank accession no. KP774835.2, 2% query coverage and 72.59% identity) and the *Erwinia* phage Pecta (GenBank accession no. MZ333131.1, 7% query coverage and 72.83% identity) [34,35]. A more detailed comparative analysis of these genomes revealed that regions of the genome with high degrees of identity (>70%) encode proteins important for phage morphogenesis: terminase large subunit, putative portal protein, tubular tail A, and major capsid protein (Figure 2).

To confirm the phylogenetic affiliation of the vB_KpnP_Klyazma phage, a proteomic tree was inferred using genomes of typical phages recommended by the International Committee on Taxonomy of Viruses (ICTV) (Figure 3). The constructed phylogeny showed that the vB_KpnP_Klyazma phage is clustered with the *Zobellviridae* family and is close to the CVT22 phage. The aforementioned was also confirmed by determining intergenomic nucleotide similarities between the members of the *Zobellviridae* family (Appendix A).

### 2.3. Host Range of vB_KpnP_Klyazma and Its Depolymerase Activity

A collection of *K. pneumoniae* strains (*n* = 180) was used to determine the host range of the vB_KpnP_Klyazma phage. According to the results of *wzi* gene sequencing, the strains had 34 unique capsular types (Appendix A). The most common capsule types in the collection comprised: KL2 (19.4%), KL23 (9.4%), KL39 (8.9%), KL64 (8.9%), and KL20 (6.1%).

vB_KpnP_Klyazma had a narrow spectrum, limited to strains with capsular type KL20 (11/11) (Table 1). However, according to the efficiency of plating (EOP) results, the phage formed separable colonies only on the host L2-1B strain, and for the remaining strains the phage only demonstrated lysis from without.

To confirm the function of the tail fibril protein (orf001), the activity of the recombinant Kl-dep depolymerase was tested on *K. pneumoniae* isolates (*n* = 11) with KL20 capsular type and four bacteriophage-insensitive control isolates (KL2, KL19, KL62, KL107). The recombinant protein formed a translucent spot on all strains with KL20 capsular type and showed no activity against control strains (Table 1). Furthermore, it is evident that the effectiveness of recombinant depolymerase on sensitive strains is dose-dependent (Figure 4). The minimum halo-forming concentration was estimated on the host strain L2-1B and amounted to 7 ng.

## 3. Discussion

All currently known *K. pneumoniae* phages belong to the class *Caudoviricetes*. The current classification by the ICTV identifies 8 families and 29 genera that include *K. pneumoniae* phages, with 7 genera currently being excluded from existing families. However, not a single *K. pneumoniae* phage from the *Zobellviridae* family has been described so far [36]. Moreover, only two phages of this family have been described to date—the *Citrobacter* phage CVT22 and *Erwinia* phage Pecta. A bacterium from the *Enterobacteriaceae* family is a host for both of the aforementioned phages [34,35].

The isolated vB_KpnP_Klyazma phage has a morphology and genome structure typical of the *Zobellviridae* family [37,38,39]. Prior to being distinguished as the *Zobellviridae* family, such phages were grouped into the T7-like supergroup due to the virion morphology and homology of structural proteins [40]. Contrary to the T7-like phages, the *Zobellviridae* family has distinct clusters of structural genes and genes that are responsible for replication, regulation, transcription, and translation. These genes are spread out across different chains. Similar division into two clusters encoded on different chains has been described for the *Escherichia* phage phiEco32 genome, which is located adjacent to the *Zobellviridae* family on the phylogenetic tree (Figure 3) [41]. Also, the *Zobellviridae* genome does not have its own RNA polymerase, which means that phage gene transcription is regulated differently. The phiEco32 phage also does not encode its own RNA polymerase, and transcription of phage genes is regulated by changing σ-factors [41].

vB_KpnP_Klyazma was found to have some differences on the genomic level compared with other members of the *Zobellviridae* family. For most phages of the family, terminal repeats are either not described or are relatively short (74–138 bp) [37,42,43], while the vB_KpnP_Klyazma phage has 711 bp long terminal repeats. During verification of terminal repeats by the Sanger sequencing method, we noticed a big drop in the intensity of the peaks on the electrophoregram during the transition from a unique part of the genome to a repeat, especially from the right end. A similar pattern was described for the phiEco32 phage [41]. These data suggest that the vB_KpnP_Klyazma phage may have a packing strategy without repeat regeneration, similar to the N4 phage. However, when considering the phylogeny inferred using the amino acid sequences of the terminase large subunit, one can observe that *Zobellviridae* phages are phylogenetically close to T1- and T7-like phages while the N4-like clade is farthest away from them. The aforementioned shows that vB_KpnP_Klyazma has a packaging type more similar to T7-like, while logically we cannot rule out a strategy with the loss of half of the synthesized DNA (Appendix A).

The vB_KpnP_Klyazma phage encodes one receptor-binding protein that is a polysaccharide depolymerase specific to *K. pneumoniae* KL20 capsular type. The phage showed productive lysis only on the host strain that belonged to ST268. In general, it should be noted that strains of these capsular and sequence types often cause serious infections, including those associated with hypervirulence and antibiotic resistance [44,45,46,47]. Three out of four KL20 and ST268 capsular type strains in the collection demonstrated a hypervirulent phenotype in string tests, although all strains had a set of genetic determinants of hypervirulence (*rmpA*/*rmpA2*, *peg-344*, *iutA*) (Appendix A). Multidrug resistance was shown by three out of four strains, which indicates the presence of convergent clones that have multidrug resistance and markers of hypervirulence.

To assess the activity of vB_KpnP_Klyazma’s depolymerase, we obtained a recombinant protein that showed effectiveness on all 11 strains of the KL20 capsular type. It is worth pointing out that the Kl-dep depolymerase formed a halo even on strains on which the vB_KpnP_Klyazma phage showed lysis from without. This is quite significant since the specificity of the depolymerase is more often the same or even lower than the host range of the phage [48,49,50]. However, the specificity of this depolymerase was quite high, because it had no effect on control strains even with capsule types containing linkages between monosugars similar to KL20 (KL19(Gal β(1–3)GlcA) and KL62(Man α(1–3)Gal)). Minimal halo-forming activity of Kl-dep was estimated at 7 ng, which is slightly higher than the usual values of 1–2 ng [48,51].

In terms of prospects for the therapeutic application of recombinant depolymerases, to date it has been shown that depolymerases can be of similar effectiveness or even more effective than phages on their own [48,49,51,52,53]. Even though depolymerases do not kill the bacteria directly, but only make the bacteria more sensitive to the action of some immune factors and some antibiotics [54], this therapy has a lot of promise because it is easier to standardize it within the current legal pharmacological standards, and such therapy has a clearer concentration-dependent mechanism of action [18,20,55].

## 4. Materials and Methods

### 4.1. Collection of K. pneumoniae Clinical Strains

A total of *n* = 180 *K. pneumoniae* clinical isolates collected throughout 2019–2022 were used in this study, *n* = 78 of which were from Raisa Gorbacheva Memorial Research Institute of Children Oncology, Hematology and Transplantation (St. Petersburg, Russia), *n* = 50 from Clinical Hospital No. 123 (Odintsovo, Russia), *n* = 40 from N.V. Sklifosovsky Research Institute of Emergency Medicine (Moscow, Russia), and *n* = 12 kindly provided by State Collection of Pathogenic Microorganisms and Cell Cultures, SCPM-Obolensk (State Research Center for Applied Microbiology and Biotechnology, Obolensk, Russia).

The bacterial strains were cultured using lysogeny broth (LB) medium (Himedia, Mumbai, India) at 37 °C. Bacterial identification was performed by MALDI-TOF mass spectrometry as described previously [56]. The antibiotic susceptibility was tested using a microdilution method according to Clinical and Laboratory Standards Institute guidelines 28th edition [57]. String test was performed as described previously [58].

### 4.2. Phage Isolation and Purification

Phage vB_KpnP_Klyazma was isolated on *K. pneumoniae* L2-1B strain in 2021 from the Klyazma River in Korolev, Moscow Oblast by a standard enrichment method [59]. Briefly, the river water sample was centrifuged at 3500× *g* for 10 min. Subsequently, the supernatant was filtered through a 0.22 µm filter (Millipore, Burlington, MA, USA). Equal aliquots (15 mL) of filtered water and double-strength LB broth were mixed with 20 μL of *K. pneumoniae* L2-1B culture at the mid-log phase (OD_620_ = 0.3). Subsequently, the mixture was incubated at 37 °C for 18 h. After that, the bacteria–phage mixture was centrifuged at 3500× *g* for 10 min to spin down most bacteria. The supernatant was then filtrated through a 0.2 μm filter (Millipore, USA). The obtained lysate was serially diluted in LB and spotted on double-layer agar plates of the host strain for phage detection and isolation. Three rounds of single plaque purification and re-infection of exponentially growing host strain yielded pure bacteriophage suspensions. Phage titers were determined using a double-agar overlay plaque assay [60].

### 4.3. Electron Microscopy of Phage Particles

TEM was used to observe the phage lysate. Carbon-coated TEM grids (Ted Pella, Redding, CA, USA) were treated for 45 s using an Emitech K100X glow discharge device (Quorum Technologies, Lewes, UK) to make the carbon surface hydrophilic and increase the adsorption of the phages. The sample was deposited onto the grids for ~1 min, stained with 1% uranyl acetate, and dried. Images were obtained using a JEM-1400 transmission electron microscope (Jeol, Tokyo, Japan) operating at 120 kV.

### 4.4. Host Range Determination

The host range of the lytic phages was established by performing the spot test [60]. Briefly, 5 mL of molten 0.7% LB agar containing 100 µL of each test bacterial culture was overlaid on 1.5% LB agar plates. Subsequently, a drop of 5 µL of vB_KpnP_Klyazma phage with titer of 10^9^ PFU/mL was spotted over LB plates. Phage resistance and susceptibility were determined by the formation of clear plaques after overnight culture at 37 °C.

Additionally, the EOP assay was performed for phage-sensitive strains as previously described [61]. The EOP value was estimated as the average plaque-forming units (PFU) on target bacteria divided by the average PFU on host bacteria and was classified according to Mirzaei and Nilsson: high productivity (EOP ≥ 0.5), medium productivity (0.1 ≤ EOP < 0.5), low productivity (0.001 < EOP < 0.1), or inefficient (EOP ≤ 0.001) [62]. If the phage did not produce single plaques, but instead formed a halo on the surface of the petri dishes, which disappeared with dilution, we called this phenomenon lysis from without.

### 4.5. Biological Characterization of vB_KpnP_Klyazma

A one-step growth experiment was performed as previously described [48]. Briefly, the mid-log growth phase culture of *K. pneumoniae* L2-1B (OD_620_ = 0.3) was infected with vB_KpnP_Klyazma at a multiplicity of infection (MOI) of 0.01 and allowed to adsorb for 10 min at 37 °C. The mixture was washed with LB medium to remove unabsorbed phages and avoid secondary adsorption. The culture was incubated at 37 °C with shaking, and samples were collected at 10 min intervals. Samples were treated with 2% chloroform, shaken briefly, and set aside for 10 min at room temperature (RT). The phage titers were then determined using the double-layer agar method. Burst size was detected by determining the ratio of the mean of virions released after bacterial infection to the mean of virions used at the beginning of the host infection.

To determine the time of adsorption of vB_KpnP_Klyazma, the host strain at the mid-log phase (OD_620_ = 0.3) was mixed with bacteriophage at a MOI = 0.01. Every 1 min from 1 to 15 min, aliquots were taken and treated with 2% chloroform, shaken briefly, and set aside for 10 min at RT. The titers of free phage were quantified by plaque assay. The following day, PFU were enumerated and the percentage of free phages was determined [63]. Both experiments of one-step growth curve and adsorption rate were conducted in triplicate.

To examine thermal stability and pH sensitivity, phage suspensions (10^11^ PFU/mL) were incubated at different temperatures (4 °C, 37 °C, 45 °C, 55 °C, 65 °C, 75 °C, and 85 °C) and pH levels (2, 3, 4, 5, 6, 7, 8, 9, 10, 11, and 12) for 1 h. The titers of phage were quantified by plaque assay. The experiment was conducted in triplicate.

### 4.6. DNA Sequencing and Genome Analysis

Bacterial DNA was extracted by using the DNA Express set (LyTeh, Moscow, Russia). Multilocus sequence typing (MLST) of *K. pneumoniae* strains was performed by determining the nucleotide sequences of seven housekeeping genes as described previously [64]. The capsular type was determined by *wzi* gene sequencing [65]. Hypervirulence-associated genes *rmpA*, *rmpA2*, *peg-344*, *fimH*, *iutA*, and *iroB* were amplified using previously published sets of primers [28,66] (Appendix A).

Extraction of the phage genomic DNA was performed using a standard phenol–chloroform extraction protocol [67]. Phage genome was sequenced using a high-throughput Illumina HiSeq system. SPAdes v3.14.0 software was used for genome assembly [68]. Phage terminal repeats were predicted with the PhageTerm tool v3.0.1 [69] and determined by direct Sanger sequencing using primers reported in Appendix A. GeneMarkS v4.32 was used for identification of open reading frames (ORFs) within the genome [70]. The tRNA genes were searched using tRNAScan-SE v2.0 [71] and ARAGORN v1.2.41 [72]. The Clinker tool was used in comparative genomic analysis [73].

Annotation of predicted genes was conducted manually using BLASTp v2.13.0, HHPred, PHROG v4, and InterPro v5.59-91.0. The absence of potentially toxic genes and antibiotic resistance determinants was confirmed by comparison with the databases Virulence Factors of Pathogenic Bacteria [74] and Antibiotic Resistance Genes Databases [75]. The annotated genome sequence of the vB_KpnP_Klyazma phage was deposited in the NCBI GenBank database under accession number OP125547.1.

### 4.7. Phylogenetic Analysis

A dataset comprising *n* = 250 complete phage genomes was retrieved from the NCBI RefSeq/GenBank database using ncbi-acc-download v.0.2.8 [76]. A tree based on pairwise distances between phage genomes was inferred using a standalone version of ViPTree v1.1.2 [77].

The terminase large subunit sequences were obtained using NCBI E-utilities. The sequences were aligned using MAFFT v7.490 and trimmed using TrimAL v1.4.rev15 with the -*gappyout* option. The phylogeny was inferred using IQ-TREE 2 v2.2.0.3 [78] with 1000 iterations of ultrafast bootstrap (UFBoot) [79] and the Q.pfam+R3 model determined by ModelFinder [80].

All trees were midpoint rooted and visualized in RStudio v2022.02.3 with R v4.1.2 using ggtree v3.2.1 [81], ggtreeExtra v1.4.2 [82], ggplot2 v3.3.5 [83], ggnewscale v0.4.7 [84], ggstar v1.0.3 [85], gginnards v0.1.1 [86], qualpalr v0.4.3 [87], and here v1.0.1 [88] packages.

### 4.8. Cloning of Coding Sequence of Depolymerase

The coding sequence of the depolymerase was amplified by using Q5 HF 2X Master mix (New England BioLabs, Ipswich, MA, USA) by primers with endonuclease restriction sites (Appendix A). PCR products were digested by *NdeI*, *XhoI* (New England BioLabs) and cloned into expression vector pGD with an N-terminal histidine tag. Competent *E. coli* Turbo cells (New England BioLabs) were transformed for plasmid propagation and isolation with subsequent Sanger sequencing. Correct constructs were used for *E. coli* T7 Express *lysY/Iq* cells (New England BioLabs) transformation followed by protein expression.

### 4.9. Protein Expression and Purification

Recombinant *E. coli* T7 Express *lysY/Iq* cells were grown in LB medium supplemented with kanamycin at 37 °C with agitation (180 rpm) until the optical density at 600 nm reached 0.5. The culture was induced by adding 0.1 mM IPTG, followed by 18 h induction at 16 °C with agitation (180 rpm). Cultures were pelleted by centrifugation (3500× *g*, 10 min, 4 °C); resuspended in His-binding buffer (His-Spin Protein Miniprep, Zymo Research, Irvine, CA, USA); lysed by 3 cycles of freeze-thawing with subsequent addition of 2.6 mg Lysozyme (Sigma-Aldrich, St. Louis, MO, USA), 1.2 U DNase I (Zymo Research), and 0.6 U RNAse A (Zymo Research); and lysed by 1 h incubation at 37 °C with agitation (250 rpm). Proteins were purified by using His-Spin Protein Miniprep (Zymo Research) according to the manufacturer’s protocol. The quality of purified protein was analyzed by SDS-PAGE and Coomassie blue staining (Appendix A).

## 5. Conclusions

In this study, we isolated and characterized a novel phage that is capable of infecting and lysing *K. pneumoniae*. The phage was a podovirus with a head size of approximately 66 nm and an 18 nm tail. Based on phylogenetic analysis, vB_KpnP_Klyazma belongs to the family *Zobellviridae*, even though its identity with the closest phages according to BLASTn was only 5%. The size and genome structure of vB_KpnP_Klyazma were similar to other described phages of the family, but the phage had a significantly larger terminal repeat of 711 bp. The phage showed productive lysis only on the host strain, but it also lysed other *K. pneumoniae* strains with the KL20 capsule type with lower efficacy. The bacteriophage receptor-binding protein vB_KpnP_Klyazma was obtained as a recombinant protein and showed depolymerase activity against all tested *K. pneumoniae* strains with KL20 capsule type.

## Figures and Tables

**Figure 1 ijms-24-04038-f001:**
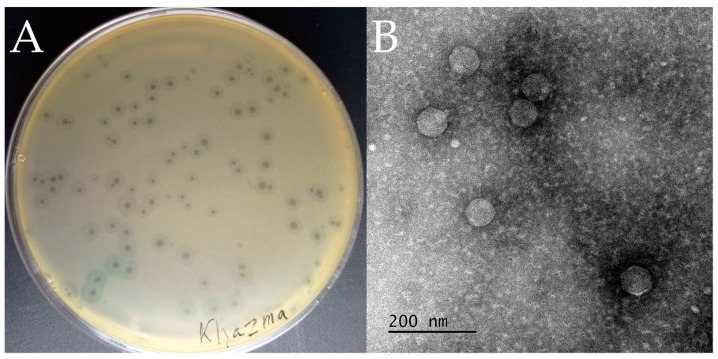
Morphological characterization of vB_KpnP_Klyazma phage. (**A**) Phage plaques on the lawn of *K. pneumoniae* L2-1B. (**B**) Transmission electron microscopy of bacteriophage particles.

**Figure 2 ijms-24-04038-f002:**
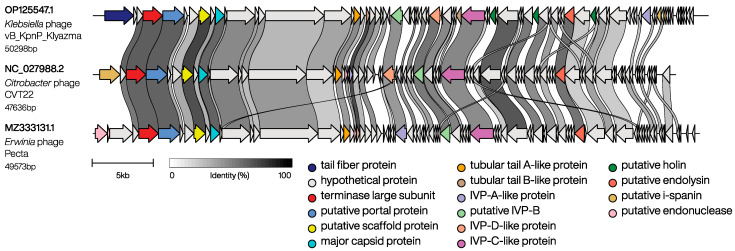
Pairwise genome comparison of phages vB_KpnP_Klyazma, CVT22, and Pecta.

**Figure 3 ijms-24-04038-f003:**
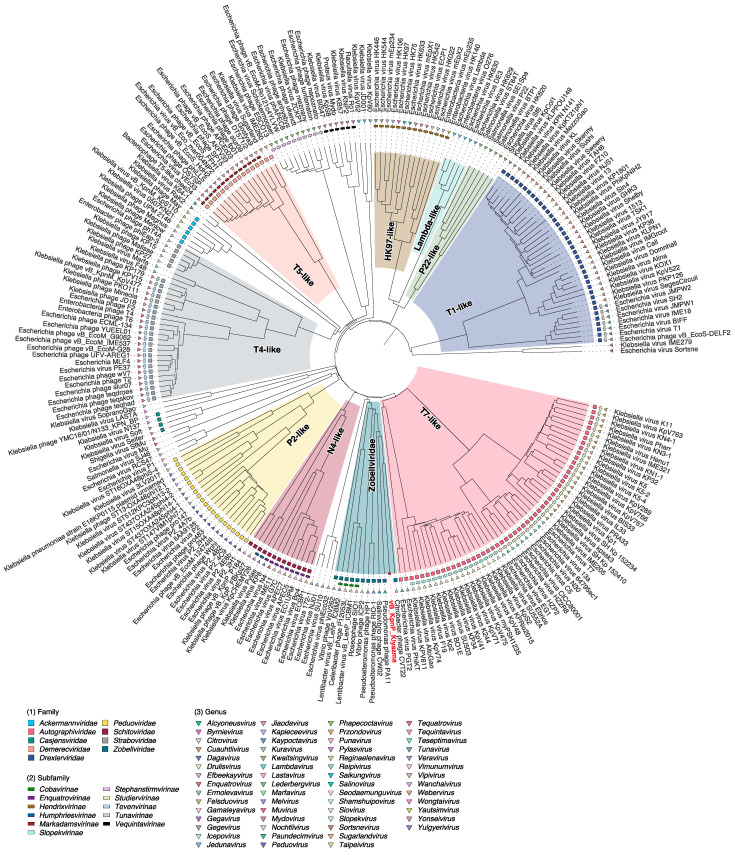
ViPTree analysis of vB_KpnP_Klyazma (highlighted in red) and related phages. Phages are identified according to their official ICTV classification.

**Figure 4 ijms-24-04038-f004:**
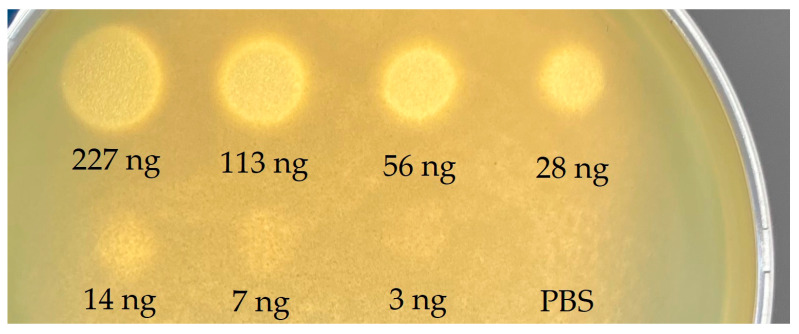
Activity of serially diluted depolymerase Kl-dep on the L2-1B host strain.

**Table 1 ijms-24-04038-t001:** Characterization of *K. pneumoniae* strains and the activity of the vB_KpnP_Klyazma phage and its recombinant depolymerase.

Strain	MLST	CPS Type	Meropenem Resistance	String Test	hvKp Genes	EOP	Kl-Dep Activity
L2-1B	268	KL20	S	+	+	highly productive	+
Kl1886	268	KL20	S	+	+	inefficient	+
Kp-25-1	268	KL20	R	+	+	lysis from without	+
KL1877	268	KL20	R	−	+	lysis from without	+
Kp-G2-6	147	KL20	S	−	−	lysis from without	+
Kp1977	147	KL20	R	−	+	lysis from without	+
KBKp7	147	KL20	R	−	−	lysis from without	+
KL2071	147	KL20	R	−	−	lysis from without	+
KL1909	147	KL20	R	−	−	lysis from without	+
Kp2307	147	KL20	R	−	+	lysis from without	+
Kp9	1544	KL20	S	+	+	lysis from without	+
Kp-40	no data	KL2	S	−	no data	no activity	−
Kp-28p	no data	KL19	R	−	no data	no activity	−
Kp2432	no data	KL62	R	−	no data	no activity	−
Kp2066	no data	KL107	R	−	no data	no activity	−

## Data Availability

The annotated complete genome of vB_KpnP_Klyazma was deposited in GenBank under accession number OP125547.1.

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
