# Peer review of "Isolation and Characterization of the First Zobellviridae Family Bacteriophage Infecting Klebsiella pneumoniae"

_ijms, 2023, doi:10.3390/ijms24044038_

Round 1
Reviewer 1 Report
This paper reports the characterization of a new type of bacteriophage that infects Klebsiella pneumonia. The paper is well written and presents new information that is of interest to the scientists who work with phages in general and to those interested in the potential to combat infections caused by this pathogen. The paper describes characteristics of the phage (morphology, host range, one step growth) as well as a detailed genetic characterization based on sequence analysis. The phage depolymerase was expressed recombinantly and some preliminary data on its properties are presented
The paper does not describe hypothesis-driven investigations.
The weakness is the lack of quantitative data on the depolymerase. The legend to Figure 4 requires further information to be understandable. Enzymatic activity should be expressed in terms of kinetics - rate of reaction etc. The only information presented is qualitative eg “high efficiency” for the type strain and a wider efficiency for other strains compared to phage lysis. This is not adequate
Author Response
Comments and Suggestions for Authors
This paper reports the characterization of a new type of bacteriophage that infects Klebsiella pneumonia. The paper is well written and presents new information that is of interest to the scientists who work with phages in general and to those interested in the potential to combat infections caused by this pathogen. The paper describes characteristics of the phage (morphology, host range, one step growth) as well as a detailed genetic characterization based on sequence analysis. The phage depolymerase was expressed recombinantly and some preliminary data on its properties are presented
We thank the reviewer for their feedback. The line numbers we refer to below are taken from the “marked up” version of the manuscript we have submitted (marked-up_manuscript.docx).
The paper does not describe hypothesis-driven investigations.
We agree with the reviewer that the work does not describe hypothesis-driven investigations. However, it should be noted that the direction of phage therapy is mostly descriptive and is still in the stage of data accumulation. From this point of view, our work is notable, as it describes the first Klebsiella bacteriophage of the Zobellviridae family. Moreover, we have shown the effectiveness of the recombinant protein, which is also a possible vector for the development of therapy.
The weakness is the lack of quantitative data on the depolymerase. The legend to Figure 4 requires further information to be understandable. Enzymatic activity should be expressed in terms of kinetics - rate of reaction etc. The only information presented is qualitative eg “high efficiency” for the type strain and a wider efficiency for other strains compared to phage lysis. This is not adequate
We thank the reviewer for bringing this issue to our attention. We found the quantification of recombinant polysaccharide depolymerases to be redundant for the purposes of this research. We were guided by the existing practice of describing phages and their depolymerases [Domingo-Calap P. et al. International journal of molecular sciences. – 2020; Chen X. et al. Virus Research. – 2022; Balcão V. M. et al. I Pharmaceutics. – 2022; Li P. et al. Pharmaceutics. – 2022; Hua Y. et al. Frontiers in Microbiology. – 2022;]. In the aforementioned publications, the enzymatic kinetics of depolymerases is not described and depolymerase activity is given in qualitative terms. Moreover, to date, articles on the therapeutic efficacy of recombinant depolymerases on animal models are well-known, but there is a lack of information about whether the kinetic characteristics of enzymes are linked to therapeutic success.
However, if you consider the issue to be relevant to this work, we are willing to perform the necessary experiments, but this would require additional time.
Furthermore, to clarify the qualitative assessments, we have added sentences to results and discussion sections (Lines 179 and 234-251); to allow a more specific assessment of the action of our depolymerase: we have revised figure 4, we have added the minimum halo forming concentration of the recombinant protein, and also compared it with other papers.

Reviewer 2 Report
Global comments
The work presented by Gorodnichevand co-workers describes the characteristic of newly isolated phage vB_KpnP_Klyazma and the effect of its depolymerase on Klebsiella strains with capsular type 170 KL20. The authors suggest that the phage vB_KpnP_Klyazma could be a promising tool for Klebsiella biocontrol.
Strengths
The introduction properly situates the subject of study. The results from the genetic characterization of phage vB_KpnP_Klyazma are interesting and correctly exposed.
Limitations
Some doubts arise in this study:
1) Lines 286-287 - authors did not write the titer of phage stock used in FOP assay, which makes it challenging to result in the interpretation.
2) in the FOP assay, the authors consider 0,0000001 as positive results. It is an extremely low result suggesting that phage vB_KpnP_Klyazma infected 9/10 strains weaker by (at least – we don't know and could only assume from results) 7 logarithmic orders. It is hard to believe that phage stock titer at lest 10^7 PFU/ml spotted on bacterial law creates only a few plaques. If they did not see single plaques they cannot say the phage infect this strain. These results rather suggest lysis from without phenomenon or phage mix.
3) Klebsiella phages and their polymerases are well characterized, as well as their activity is detailed described. Therefore results obtained in this study should be discussed with the existing literature background including the comparison of phage vB_KpnP_Klyazma and its depolymerase anti-Klebsiella activity in the Discussion section.
I include all my comments, and I hope the authors are willing to take the time to make this work a solid publication that the phage community can refer to.
I refer to the line number kindly provided by the authors.
Line 20 – I suggest canceling the information about phage head size in the Abstract section
Lines 23-24 although its identity with the 23 closest members of this family was not higher than 5% - suggested that this phage does not belong to the Zobellviridae family. Please reformulate this sentence.
Line 25 – all strains – please write the number of strains
Line 28 - The sentence should be reformulated to Recombinant depolymerase showed enhanced activity…
Line 50 – term metastatic spread is used only for cancer cells' invasion of distal tissues, not in the context of bacterial invasion and colonization. Please reformulate it.
Line 69, 144 – different fonts
Lines 81-82 – please give the link to Supplementary materials with the levels of resistance to antibiotics
Line 66 – why the fimH gene? It is a common gene in all Enterobacteriaceae family members.
Figure 4 – please give details on the strain presented in Figure 4
Line 265 – please specify the source of the water
Lines 286-287 – please write the titer of phage working stock
Author Response
Comments and Suggestions for Authors
Global comments
The work presented by Gorodnichevand co-workers describes the characteristic of newly isolated phage vB_KpnP_Klyazma and the effect of its depolymerase on Klebsiella strains with capsular type 170 KL20. The authors suggest that the phage vB_KpnP_Klyazma could be a promising tool for Klebsiella biocontrol.
Strengths
The introduction properly situates the subject of study. The results from the genetic characterization of phage vB_KpnP_Klyazma are interesting and correctly exposed.
Limitations
Thank you for the valuable comments on our manuscript. We did our best to improve the manuscript and to clarify the questions that were raised. The line numbers we refer to below are taken from the “marked up” version of the manuscript we have submitted (marked-up_manuscript.docx).
Some doubts arise in this study:
1) Lines 286-287 - authors did not write the titer of phage stock used in FOP assay, which makes it challenging to result in the interpretation.
We have added this information to the “Materials and Methods section”
Line 295: “Subsequently, a drop of 5 µl of vB_KpnP_Klyazma phage with a titer of 109 PFU/ml, was spotted over LB plates”
2) in the FOP assay, the authors consider 0,0000001 as positive results. It is an extremely low result suggesting that phage vB_KpnP_Klyazma infected 9/10 strains weaker by (at least – we don't know and could only assume from results) 7 logarithmic orders. It is hard to believe that phage stock titer at lest 10^7 PFU/ml spotted on bacterial law creates only a few plaques. If they did not see single plaques they cannot say the phage infect this strain. These results rather suggest lysis from without phenomenon or phage mix.
Thank you for catching this discrepancy. We have corrected the interpretation of the EOP assay results according to your suggestion:
Lines 170-172: «However, according to the efficiency of plating (EOP) results, the phage formed separable colonies only on the host L2-1B strain, and for the remaining strains the phage only demonstrated lysis from without.»
Similar corrections have been made to Table 1 and Supplementary Table 1. Line 183
Also, the “Materials and Methods” section has been updated as follows:
Lines 301-303: «If the phage did not produce single plaques but formed a halo on the surface of the petri dishes, which disappeared with dilution, we called this phenomenon lysis from without.»
3) Klebsiella phages and their polymerases are well characterized, as well as their activity is detailed described. Therefore results obtained in this study should be discussed with the existing literature background including the comparison of phage vB_KpnP_Klyazma and its depolymerase anti-Klebsiella activity in the Discussion section.
We have revised the “Discussion” section to include the most recent data from this area of research:
Lines 234-250: «To assess the activity of the vB_KpnP_Klyazma’s depolymerase, we have obtained a recombinant protein that showed its effectiveness on all 11 strains of the KL20 capsular type. It is worth pointing out that KL-depolymerase formed a halo even on those strains on which the vB_KpnP_Klyazma bacteriophage showed lysis from without. This is quite significant since the specificity of the depolymerase is more often the same or even lower than the host range of the bacteriophage [48–50]. However, the specificity of this depolymerase was quite high, because it had no effect on control strains even with capsule types containing similar to KL20 linkages between monosugars [KL19(Gal β(1-3)GlcA) and KL62 (Man α(1-3)Gal)]. Minimal halo-forming activity of Kl-dep was estimated as 7 ng, which is slightly higher than the usual values of 1-2 ng [48,51].
In terms of prospects for the therapeutic application of recombinant depolymerases, to date it has been shown that depolymerases can be similar or even more effective than bacteriophages on their own [48,49,51–53]. Even though depolymerases do not kill the bacteria directly, but only make the bacteria more sensitive to the action of some immune factors and some antibiotics [54], this therapy has a lot of promise because it is easier to standardize it within the current legal pharmacological standards, and such therapy has a clearer concentration-dependent mechanism of action [18,20,55]»
I include all my comments, and I hope the authors are willing to take the time to make this work a solid publication that the phage community can refer to.
I refer to the line number kindly provided by the authors.
Line 20 – I suggest canceling the information about phage head size in the Abstract section
Thank you for your suggestion. The “Abstract” section was revised accordingly in line 20
Lines 23-24 although its identity with the 23 closest members of this family was not higher than 5% - suggested that this phage does not belong to the Zobellviridae family. Please reformulate this sentence.
We would have to disagree with this statement. Due to the ongoing revision of viral phylogenetics, nucleotide identity is currently not a sufficient feature to distinguish families. We were guided by recommendations given in an article by ICTV members [Turner D., Kropinski A. M., Adriaenssens E. M. A roadmap for genome-based phage taxonomy //Viruses. - 2021. - Т. 13. - №. 3. - С. 506.]. According to these recommendations tailed phage families are distinguished on the basis of the following criteria:
- The family is represented by a cohesive and monophyletic group in the main predicted proteome-based clustering tools (ViPTree, GRAViTy dendrogram, vConTACT2 network).
- Members of the family share a significant number of orthologous genes (the number will depend on the genome sizes and number of coding sequences of members of the family), see genus section for methods.
- If a family-level cluster shares orthologues with another family-level cluster, the family cluster needs to be monophyletic in a phylogenetic analysis of the shared orthologue(s).
Line 25 – all strains – please write the number of strains
Line 28 - The sentence should be reformulated to Recombinant depolymerase showed enhanced activity…
Thank you for your remarks, we have revised these sentences:
Lines 24-25: «The bacteriophage demonstrated lytic activity against all (n=11) K. pneumoniae with the KL20 capsule type, but only the host strain was lysed effectively. »
Lines 28-30: « The ability of recombinant depolymerase to cleave bacterial capsular polysaccharides regardless of a phage's ability to successfully infect a particular strain holds promise for the perspective of using depolymerases in antimicrobial therapy, even though they only make bacteria sensitive to environmental factors, rather than killing them directly.»
Line 50 – term metastatic spread is used only for cancer cells' invasion of distal tissues, not in the context of bacterial invasion and colonization. Please reformulate it.
You are right, metastatic spread as a term commonly used to describe cancer cell invasion of distal tissues. On the other hand, some authors use this term to describe manifestation of infection to other tissues [Lindstrom S. T. Aust NZ J Med. – 1997; Fang C. T. et al. The Journal of experimental medicine. – 2004; Ma L. C. et al. The Journal of infectious diseases. – 2005; Chang D. et al. Frontiers in Microbiology. – 2021; Lan P. et al. Journal of Global Antimicrobial Resistance. – 2021; Dong N. et al. EBioMedicine. – 2022]. However, as you feel it is necessary, we have revised the corresponding sentence:
Lines 50-51: « This pathotype is able to cause severe infections in immunocompromised people and in people who were healthy prior to infection, and can lead to subsequent intra-abdominal abscesses, which are prone to manifest onto other tissues [7–9].»
Line 69, 144 – different fonts
Thank you for your comment. We have harmonized the fonts to a consistent standard
Lines 81-82 – please give the link to Supplementary materials with the levels of resistance to antibiotics
We have made the necessary refinements
Line 82 «The strain showed sensitivity to meropenem, cefotaxime, gentamicin, levofloxacin, tetracycline, colistin and was resistant to erythromycin (Supplementary Table 1)».
Line 66 – why the fimH gene? It is a common gene in all Enterobacteriaceae family members.
Indeed, fimH is a widespread gene encoding the adhesive subunit of type 1 fimbriae and can be found in most species of the family Enterobacteriaceae. However, this gene encodes one of the important adhesins essentially determining the virulence of K. pneumoniae against human cells [Stahlhut S. G. et al. Journal of bacteriology. – 2009; Zhou C. et al. Infection and Drug Resistance. – 2021; Hu D. et al. Frontiers in cellular and infection microbiology. – 2021; Bautista-Cerón A. et al. Microorganisms. – 2022]. Moreover, there is some evidence that this gene is present in 90% of all tested K. pneumoniae strains [Stahlhut S. G. et al. Journal of bacteriology. – 2009; Zhou C. et al. Infection and Drug Resistance. – 2021; ] and is also identified in 100% of hypervirulent pathotype strains [Xu M. et al. Infection and drug resistance. – 2019], allowing us to use this gene as a kind of positive control.
Figure 4 – please give details on the strain presented in Figure 4
We elaborated this by revising figure 4 (Line 186) and its caption (Line187). We have also specified the minimum halo forming concentration
Lines 179-180: «The minimum halo forming concentration was estimated on the host strain L2-1B and amounted to 7 ng»
Line 265 – please specify the source of the water
We have made the necessary revisions to the “Materials and Methods” section
Line 271 «Briefly, the river water sample was centrifugated at 3500g for 10 min.»
Lines 286-287 – please write the titer of phage working stock
Thank you for your remarks, we have made the necessary refinements to the “Materials and Methods” section
Line 295: «Subsequently, a drop of 5 µl of vB_KpnP_Klyazma phage with a titer of 109 PFU/ml was spotted over LB plates.»

Round 2
Reviewer 1 Report
no further comment
Author Response
Thank you for your valuable comments and recommendations. Thank you for making this article better.
On the advice of the editor, we changed the title of the manuscript
"Isolation and characterization of the first Zobellviridae family bacteriophage infecting Klebsiella pneumoniae"

Reviewer 2 Report
Accept in present form.
Author Response

(The authors gave the same response as above.)
